# Analysis of Plantar Tactile Sensitivity in Older Women after Conventional Proprioceptive Training and Exergame

**DOI:** 10.3390/ijerph20065033

**Published:** 2023-03-13

**Authors:** Claudio Henrique Meira Mascarenhas, José Ailton Oliveira Carneiro, Thaiza Teixeira Xavier Nobre, Ludmila Schettino, Claudineia Matos de Araujo, Luciana Araújo dos Reis, Marcos Henrique Fernandes

**Affiliations:** 1Graduate Program in Nursing and Health, Universidade Estadual do Sudoeste da Bahia (UESB), Jequié 45210-506, Brazil; 2Post-Graduate Program in Physical Education, Universidade Estadual do Sudoeste da Bahia (UESB), Jequié 45210-506, Brazil; 3PPG QUALISAÚDE: Quality in Health Services and Patient Safety, Federal University of Rio Grande do Norte, Natal 59200-000, Brazil

**Keywords:** clinical trial, aging, sensitivity, exercise, physical therapy

## Abstract

Objective: To evaluate and compare the effects of conventional proprioceptive training and games with motion monitoring on plantar tactile sensitivity in older women. Methods: A randomized controlled clinical trial, with 50 older women randomized into three groups: conventional proprioception (n = 17), games with motion monitoring (n = 16), and the control (n = 17). They underwent 24 intervention sessions, three times a week, for eight weeks. The conventional proprioception group performed exercises involving gait, balance, and proprioception. The games performed by the motion monitoring group included exercises using the Xbox Kinect One video game from Microsoft^®^. The evaluation of tactile pressure sensitivity was performed using Semmes–Weinstein monofilaments. Intragroup comparisons between the two paired samples were performed using paired Student’s *t*-test or Wilcoxon test. Intergroup comparisons between the three independent samples were performed using the Kruskal–Wallis test and Dunn’s post hoc test, with *p* ≤ 0.05. Results: The older women submitted to conventional games with motion monitoring training and showed improvement in plantar tactile sensitivity in the right and left feet. When comparing the intergroup results, the two training modalities obtained an improvement in the plantar tactile sensitivity of the older women when compared to the control group. Conclusions: We conclude that both training modalities may favor the improvement of plantar tactile sensitivity in older women, with no significant differences between conventional and virtual training.

## 1. Introduction

Aging is a natural, dynamic, and progressive process, which is accompanied by morphological, functional, and biochemical changes, and an increase in the risk of several diseases, especially those related to the feet (neuropathy, diabetes, peripheral neuropathy, plantar fasciitis, diabetic foot, and others) [1,2]. In this regard, for every three older women living in the community, at least one of them has foot problems, especially women, who have about twice as many foot problems classified as moderate and severe [3].

In the somatosensory system, the peripheral nervous system can undergo changes such as the loss of myelinated and unmyelinated fibers and a decrease in nerve conduction velocity, leading the elderly to have a deficit in sensory discrimination [4,5,6]. With aging, there is a loss of receptors and also a reduction in the number of sensory fibers that innervate the peripheral receptors, which may cause peripheral neuropathies that affect the proprioceptive system [7,8].

The changes that arise in the feet of the elderly can compromise the performance of activities of daily living, interfering negatively with posture and gait and contributing to the development of disabilities. Among these alterations is the plantar tactile sensitivity impairment, which can bring consequences such as postural instability, gait disorders such as a reduction in the balance phase, speed and step symmetry, discomfort, pain, deformities, and risk of falls, thus impairing the quality of life of the elderly [4,5,6].

Management strategies focus on prevention, early detection, and the appropriate treatment of aging-related morbidities, reducing the socioeconomic impact, not only for the individual but also for society [6,7,8]. Proprioceptive rehabilitation is a resource that is widely used in physiotherapy with the purpose of stimulating the sensorimotor system. There are many ways and tools to structure these exercise programs; among them, the training conventional proprioceptive method is widely disseminated in the literature and relies on the use of materials such as balls, sticks, balance boards, and mats, among other resources for this purpose [2,3,4,5].

Despite the relevance of the problem, there is a scarcity of studies available in the literature that is aimed at comparing the effects of different training modalities on plantar tactile sensitivity in the healthy elderly population. Some research with emphasis on proprioceptive stimulation, sensory stimulation, and active exercises have demonstrated important results in the improvement of tactile sensitivity, but these studies include older women with some clinical impairment [7,9]. From this perspective, the present study aimed to evaluate and compare the effects of conventional proprioceptive training and games with motion monitoring on the plantar tactile sensitivity of older women.

## 2. Methods

### 2.1. Characterization of the Study

This is a randomized controlled clinical trial, which was developed according to the recommendations of CONSORT (consolidated standards of reporting trials) [10]. It was developed in the city of Jequié-Bahia, and the sample was composed of older women who participated in four senior citizenship groups.

### 2.2. Ethical Issues

This study was approved by the Research Ethics Committee of the Universidade Estadual do Sudoeste da Bahia (UESB) under opinion number 2.627.047, CAAE: 46887315.1.0000.005. The study was registered in the Brazilian Registry of Clinical Trials (REBEC) database, registration number RBR-592yyp.

### 2.3. Population and Sample

To include the participants in the study, the following criteria were used: (a) a maximum age of 79 years old; (b) no practice of any type of physical exercise in the last three months; (c) absence of cognitive deficit, diabetes mellitus, vestibulopathies, cardiovascular diseases, visual, or hearing impairment; (d) independent ambulation and locomotion without auxiliary devices. The study excluded older women who had attended another proprioceptive rehabilitation program during the training or in the last three months and those who participated in less than 75% of the training program.

The sample size was defined based on the results of a pilot study with 5 older women in each group and had as an outcome the difference (i.e., performance before training or control—performance after training or control) in the TUGT (Timed Up & Go test). For the sample calculation, α = 0.05 and test power (1-β) = 0.95 were considered, with 3 groups (control x conventional x exergame), which obtained a sample number of 36 individuals (i.e., 12 in each group). Considering the possibility of sample loss over the course of the 8-week intervention, the sample size was estimated with a 25% loss margin in each group; therefore, a sample number of 15 older women per group was expected (i.e., a total sample of 45 elderly). The calculation of the sample size was performed using the G*Power^®^ software version 3.1.

After screening the participants according to the established criteria, and including 50 older women remained in the sample, which was submitted to stratified randomization by age (60–69/70–79) and BMI (low—less than or equal to 22.0/high—greater than or equal to 27.0), thus seeking a greater homogeneity in the allocation of the older women among the groups. From this stratification, the participants were distributed into four groups: age (60–69) and low BMI, age (60–69) and high BMI, age (70–79) and low BMI, and age (70–79) and high BMI.

### 2.4. Procedures

Subsequently, a code was created for each participant, and randomization was performed in blocks of three individuals for each stratum. The blocks were randomized using Microsoft Excel version 2013 software, and subsequently, the codes were distributed in three arms of the study (control group, conventional group, and exergame group). The entire process was performed by a researcher with no clinical involvement in the trial, thus ensuring allocation confidentiality.

The control and conventional groups were composed of 17 participants, and the exergame group was composed of 16 participants; at the end of the study, each group ended with 15 participants. The losses were related to participation below 75% of the training program (3 older women) and dropouts (2 older women), totaling 5 losses.

The control group (GCT), during the intervention period, did not participate in any training modality; the conventional group (GCV) participated in conventional proprioceptive training; and the exergame group (GEX) participated in proprioceptive training based on virtual realities.

#### 2.4.1. Conventional Proprioceptive Training (Conventional Group/GCV)

The training was carried out three times a week, for 8 weeks, for a total of 24 sessions, with a duration of 50 min per session and a minimum interval of 48 h between each session. The training protocol was organized as follows: warm-up (10 min), proprioceptive training (30 min), and cool-down (10 min), with the monitoring of blood pressure and heart rate before and after the activities.

The warm-up was performed with walking (4 min) and stretching exercises for the muscles of the upper and lower limbs and spine (6 min). The warm-up was conducted with breathing exercises (5 min) and stretching exercises (5 min). The participants were warned not to alter their activities of daily living during the intervention period, thus avoiding the possible influences of external factors on the outcomes of the research.

The conventional proprioceptive training protocol involved gait, balance, and proprioception training and was spatially organized in the form of a circuit with different textures and obstacles, consisting of seven stations. The materials used were: 1 dense mattress of dimension 120 × 70 × 10 cm (station 1), 1 foam module—a mini beam of dimension 190 × 22 × 10 cm (station 2), 4 agility rings with 42 cm diameter (station 3), 1 proprioceptive lateral board of dimension 60 × 36 × 8 cm (station 4), 2 agility cones of the dimensions 23 × 14 cm (station 5), 1 proprioceptive disc of 40 cm in diameter (station 6), and 3 agility barriers of the dimensions 70 × 15/ 70 × 20/ 70 × 25 cm (station 7).

The older women participated, in groups of two or three, in specific exercises at each station that were combined with sensory and motor stimulation, as follows:-Station 1: Lateral strides (right and left), forward and backward strides on an unstable surface (dense mattress), exercises in bipodal and unipodal support (right and left) with eyes open and closed, agility training with ball throwing.-Station 2: Forward, backward, and sideward march (right and left) with a narrow base on an unstable surface (mini foam board), march alternating between floor and mini-board, agility training with ball throw.-Station 3: Forward, backward, sideways, and cross-legged march between the agility rings.-Station 4: Latero-lateral and anteroposterior exercise on the lateral proprioceptive board with eyes open and closed, agility training with ball throw.-Station 5: Forward, backward, and sideward march between cones with a narrow base and circumferential path with full foot support, with heel support only, and with forefoot support only.-Station 6: Exercises on the proprioceptive disk with multidirectional shifts with eyes open and closed and agility training with ball throwing.-Station 7: Forward, backward, and sideward march over agility barriers and agility training with ball throw.

Each participant remained for two minutes at each station, with a thirty-second break between stations. After going through all seven stations, the forward, sideways, and backward march was performed again through all the stations continuously without breaks, and only a thirty-second break at the end of each circuit, until the proposed time of 30 min was completed.

The degree of difficulty was increased throughout the training through the speed and execution of the activities. In all sessions, each elderly woman was accompanied by a researcher, and the execution and physical capacity of each participant in relation to the execution of the activities were taken into consideration. The exercises of the conventional training protocol were based on the consulted literature [9,10].

#### 2.4.2. Proprioceptive Training Based on Virtual Realities (Exergame Group/GEX)

The proprioceptive training based on virtual realities “exergames” was performed using the Xbox Kinect One videogame from Microsoft^®^. This console uses technology with motion sensors and the Kinect, which captures the movements of the players, i.e., they are sensitive to changes in direction, speed, and acceleration, thus allowing the games to be controlled with body movement without the need to use any manual control [11].

The game used was Kinect Sports Rivals, which simulates six sports activities: jet ski racing, climbing, soccer, bowling, tennis, and target shooting. The selection of the games was guided by the analysis of the motor demands offered by each one of them, which ranged from basic motor skills, such as squatting and lifting, jumping, turning, tilting the trunk, moving laterally and anteroposteriorly, and moving the arms in all directions to more complex motor skills that stimulated coordination, balance, stability, and proprioception, such as extending an arm and flexing the contralateral leg, which is associated with body thrust (climbing game). This also included performing lateral-lateral displacements associated with the flexion/extension and adduction/abduction movements of the upper limbs (a tennis game); performing kicks, displacements, and body rotation (a soccer game); performing hip and knee flexion, with trunk rotation and inclination (jet ski game); and performing hip, knee, and ankle flexion with lower limbs in an alternate position, associated with trunk inclination and shoulder flexion/extension movement (bowling game).

The training with exergames was carried out in a room with no objects that could interfere with the performance of the older women, and in which the games were projected on the wall using an Epson Power Lite S8+ projector and a set of Multilaser 60 WRms Sp088 speakers. The participants were accompanied by researchers and performed the activities in pairs, barefoot, and positioned in front of the Kinect sensor at a distance of three meters.

Each session consisted of training with three games previously selected by lottery, and the duration of each game was 10 min, for a total of 30 min. The order of the games in each session was also conducted by lottery; every six sessions, a new lottery was held, where one game was replaced by another, allowing the participants to have contact at the end of the training with all five selected games.

#### 2.4.3. Instruments

The study used a questionnaire with sociodemographic variables (age, marital status, education, and monthly family income) and health-related variables (body mass index/BMI, presence of diagnosed diseases, musculoskeletal pain in the last 7 days, musculoskeletal pain in the last 12 months, and medications), and the evaluation of plantar tactile sensitivity.

The evaluation of tactile pressure sensitivity in the plantar region was performed through the Semmes-Weinstein monofilaments “esthesiometer” of the brand SORRI^®^, which are composed of six nylon filaments of equal length of different colors and various diameters that produce a standardized pressure on the skin surface.

The monofilaments have the purpose of evaluating and quantifying the tactile perception threshold and sensation of deep foot pressure [11]. The classification of the filaments is based on their colors, as follows: green color (0.05 gf) and blue color (0.2 gf): normal sensitivity; violet color (2.0 gf): difficulty with shape and temperature discrimination; dark red color (4.0 gf): mild loss of protective sensation, vulnerable to injury; orange color (10.0 gf): mild loss of protective sensation; magenta color (300.0 gf): loss of protective sensation; no response: total loss of sensitivity.

The monofilaments were applied to 10 different points on each foot, as predefined by Armstrong et al. [12], consisting of the plantar region (PR) of the 1st toe; PR of the 3rd toe; PR of the 5th toe; PR of the 1st metatarsal; PR of the 3rd metatarsal; PR of the 5th metatarsal; the medial region (MR) of the plantar surface of the foot; the medial-lateral region (MLR) of the plantar surface of the foot; calcaneus; and the interphalangeal region (IR) between the 1st and 2nd toe. The evaluation protocol followed the instructions in the user’s manual of the manufacturer of the product “SORRI^®^ Esthesiometer”, as well as other studies [13,14] (Figure 1).

Before starting the procedure, a test was performed with the monofilament, which was applied to an area of the participants’ arm with preserved sensitivity so that the correct understanding of the test could be verified. The participants were positioned on a stretcher in a supine position, eyes closed, and in a quiet environment. Each monofilament was applied perpendicularly for about 1–2 s at each point so as to curve over the area without sliding over the skin of the elderly woman. The tests started with the thinnest and lowest pressure monofilament (0.05 gf, green color), and in case of no response, a monofilament of larger diameter and pressure (0.2 gf, blue color) was used, and so on until the participant was able to detect the touch.

Each monofilament was pressed onto the skin, and the participant was instructed to indicate the time and place when she felt the pressure of the filament. The application was repeated twice on the same site and alternated with at least one simulated application in which the monofilament was not applied. This way, three questions were asked per application site and were considered an absent sensation if two answers were incorrect in the three attempts. It is noteworthy that the elderly were strictly monitored over the 8 weeks, noting their non-participation in activities that could influence the study.

The recording of the tests was made by marking at each established point the color corresponding to the first monofilament that the participant detected by touch. To allow a comparison between the situations, a numerical score was stipulated for each monofilament that ranged from 0 (zero) no perception to 6 (six) normal sensitivity; that is, the higher the score, the better the plantar tactile sensitivity. The sensitivity was determined by regions of the right and left feet: the forefoot (the sum of the points of seven regions), the midfoot (the sum of the points of two regions), the hindfoot (the score of one region), and the whole foot (the sum of all points assessed).

The evaluations of the variables were carried out in two moments: before training (T0) and after training (T1), by researchers who did not participate in the allocation process of older women and had no contact with the treatment groups. For the CGT, the participants were evaluated and reassessed following the same period and place established for the intervention groups.

### 2.5. Data Analysis

To evaluate the homogeneous behavior of quantitative variables (age and BMI) at the baseline in the three groups (control, conventional, and exergame), the analysis of variance (ANOVA) and Kruskal–Wallis tests were used after checking the normality of the data using the Shapiro–Wilk test. Pearson’s chi-square test and Fischer’s exact test were used to comparing the categorical variables (marital status, education, family income, presence of diseases, pain in the last 7 days and 12 months, and medications) between the groups at the beginning of the study.

In the inferential analysis (parametric or nonparametric) for the comparisons of plantar tactile sensitivity variables, the Shapiro–Wilk test was initially used to test the normality of the data. Intragroup comparisons between the two paired samples were performed using paired Student’s t-test or Wilcoxon test. Intergroup comparisons between three independent samples were performed using the Kruskal–Wallis test, and, in the case of statistical difference, Dunn’s post hoc test was used.

The effect size was calculated for between-group comparisons (i.e., comparisons of differences between T0 and T1) using the partial eta2 parameter (partia leta squared, η^2^ partial) as an effect size indicator. The interpretation of the effect size adopted a small effect size when η^2^ = 0.01, a medium effect size when η^2^ = 0.06, and a large effect size when η^2^ = 0.14. The significance level adopted in all analyses was 5% (α = 0.05), and the data were analyzed in an IBM Statistical Package for the Social Sciences (SPSS) for Windows, version 21.0.

## 3. Results

The analysis of plantar tactile sensitivity, specifically in the forefoot, midfoot, hindfoot, and right and left whole foot regions in the control, conventional, and exergame groups at T0 showed that the variables did not show significant differences between the groups, indicating that the three groups had similar characteristics at the baseline of the study (Table 1).

The comparisons between T0 and T1 in the control group showed a significant difference in the tactile sensitivity of the hindfoot R, midfoot L, hindfoot L, and whole foot L, indicating that, at the end of the period evaluated, the older women in this group presented significantly lower values in these regions, which characterized a worsening of the plantar tactile sensitivity (Table 2).

The comparisons between T0 and T1 in the conventional group showed a significant difference in all variables, except for the tactile sensitivity of the forefoot R, indicating that, at the end of the treatment, the older women in this group presented significantly higher values, characterizing an improvement in the plantar tactile sensitivity of both feet, especially in the left foot (Table 3).

The comparisons between T0 and T1 in the exergame group showed a difference in the tactile sensitivity of the forefoot R, whole foot R, forefoot L, and whole foot L, indicating that, at the end of the treatment, the older women in this group presented significantly higher values in these regions, which characterized an improvement in the plantar tactile sensitivity of both feet (Table 4).

The comparative analysis of the changes in the plantar tactile sensitivity variables showed significant differences between the groups. For the tactile sensitivity of the hindfoot R, whole foot R, forefoot L, hindfoot L, and whole foot L, the results showed a better effect of conventional and exergame training when compared to the control group. Regarding the sensitivity of the forefoot R, there was a better effect in the exergame training compared to the control group. As for the sensitivity of the midfoot R and L, there was a better effect in conventional training compared to the control group (Table 5).

Among all the variables studied, no significant differences were observed between the conventional and exergame groups, indicating a similar effect of the two types of training on the plantar tactile sensitivity of both feet. Regarding the effect size, the results indicate an effect that is classified as large for all the variables studied (0.141–0.295) (Table 5).

## 4. Discussion

The results of the present study demonstrated that the older women from the control group, i.e., those who did not undergo any kind of intervention during the studied period, presented a worsening of the plantar tactile sensitivity in both feet, with predominance in the left foot.

The impairment of the plantar tactile sensitivity that was observed in the older women of the control group may have made them more susceptible to suffering falls or to present difficulties in locomotion on uneven surfaces so that, according to Cenci et al. [15], the decrease in plantar sensitivity was one of the main factors that collaborated to the decrease in afferences to the motor control system, thus generating a decrease in balance, impairment of gaits, such as smaller cadence, shorter steps, and less acceleration, slowness in the correction of motor errors and obstacle transposition.

In relation to the older women submitted to proprioceptive conventional and games with motion monitoring training, an improvement in the plantar tactile sensitivity of the right and left feet was evidenced in both groups. The improvement of this sensitivity can be attributed to the multisensory stimuli provided by proprioceptive training since studies have shown that physical exercises that promote stimuli and vary in texture, weight, and shape, whether associated or not with sound and visual stimuli, improve the blood supply to the lower limbs, thus contributing to the reduction in endoneural hypoxia and an improvement in nerve conduction [8,14,15].

Santos et al. [16] investigated the effect of conventional proprioceptive training on plantar tactile sensitivity in sedentary women. The study used a conventional proprioceptive protocol that was similar to the one developed in the present study, which involved gait, balance, and proprioception training in order to provide sensory stimulation on the plantar surface. The results showed significant improvement in plantar sensitivity after 24 intervention sessions. However, it is worth mentioning that most of the participants were younger than 60 years old, ranging from 50 to 70 years old; and that the results were not compared to other types of training, which made it impossible to generalize and compare the effects with another intervention.

Studies have concluded that orthostasis on textured surfaces with varying densities caused an increase in peripheral nerve activity for healthy individuals as a function of changes in the transmission of afferent signals from the sole of the foot [13,14,15,16]. In the present study, the conventional training program adopted used techniques and resources similar to those addressed by the aforementioned authors. This variety of techniques and resources may have contributed to the stimulation of different areas of the foot innervated by the deep peroneal, sural, saphenous, and tibial nerves as well as to the activation of a greater quantity of exteroceptors, which provided an improvement of tactile sensitivity in the distinct plantar regions of older women who underwent this type of training.

Regarding training with games and motion monitoring (GMM), although no studies have been found on its effects on plantar tactile sensitivity, the present study showed that this training modality contributed to an improvement in sensitivity. This resource consists of the reproduction of tasks that can be performed by the individual in interaction with a multidimensional and multisensory environment created by a computer that can be explored in real-time, thus contributing to sensory stimulation [17,18].

One possible explanation for the improvement in plantar tactile sensitivity from VR is that this type of training is able to provide tactile feedback to the central nervous system. The additional sensory stimulation of plantar cutaneous receptors improves tactile sensitivity, favoring the performance of activities and preventing risks of accidents since the perception of movement is favored when tactile feedback is available [18].

According to some authors, training based on VR allows for a greater number of repetitions, high variability of movements, and auditory and visual feedback [18,19,20]. All this range of activities provided by training with VR ends up exerting a positive influence on plantar tactile sensitivity. However, unlike the conventional training, which was developed on unstable surfaces and with different textures and densities, the VR was performed only on the ground, that is, on a stable surface and without different textures and densities, which may not have favored the stimulation and, consequently, the improvement of tactile sensitivity in all specific regions of the feet of this group of older women.

The results of the present study showed that when the three groups of older women were compared, the plantar tactile perception presented significant differences when analyzing the before and after interventions, in which the conventional and exergame groups obtained better effects when compared to the control group. However, no significant differences were observed in plantar tactile sensitivity between the conventional and exergame groups.

Based on these results, it is possible to state that the improvement of plantar tactile sensitivity in older women could be achieved with a physiotherapeutic intervention of easy access and low cost, as in the case of conventional training. The proposal of training, with games and motion monitoring, must be considered a great advance in the health area, since in the globalized world, participation in the technological process is inevitable, and the use of these advances by the health sciences is undeniable.

Although VR is a resource that has higher costs than conventional physical therapy, it has shown positive results in several clinical areas. Similar to conventional training, VR is a therapeutic modality that can be performed in physical therapy clinics, in long-stay institutions for the elderly, or even at home, facilitating the treatment of people who do not have access to a rehabilitation center. In view of all the findings, both types of training must complement each other, always seeking the physical and emotional well-being of the elderly population.

It is important to emphasize the need for further studies that can describe more characteristics and factors that are related to the effects of different exercise modalities on the sensory and functional responses of older women since this was one of the limitations of the present study.

Another limitation of the present study was that a follow-up was not carried out to verify how long the effects of the intervention lasted after its completion. Thus, further studies are suggested, given the importance of gaining a greater understanding of this subject for possible reference measures and an improvement in the quality of life of this population.

## 5. Conclusions

Based on the results of the present study, it is suggested that the proposed proprioceptive training with conventional games and motion monitoring may be favorable for the improvement of plantar tactile sensitivity in older women. When comparing the intergroup results, there was a better effect of the intervention groups when compared to the control group; however, this was without significant differences between conventional and virtual training regarding the outcomes studied.

## Figures and Tables

**Figure 1 ijerph-20-05033-f001:**
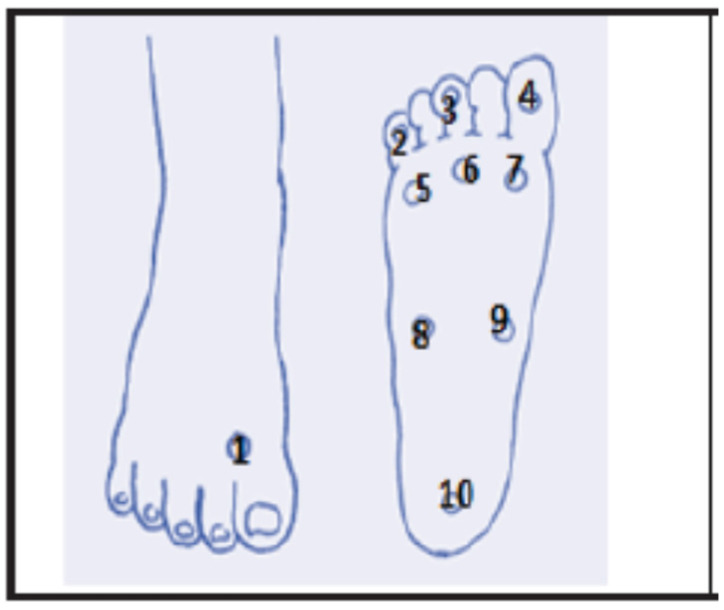
Points for assessing plantar sensitivity. Forefoot (1 to 7), midfoot (8 and 9), hindfoot (10).

**Table 1 ijerph-20-05033-t001:** Intergroup comparisons of the initial plantar tactile sensitivity (T0) of the older women participating in the study. Jequié, Bahia, 2022.

Variable	Control	Conventional	Exergame	*p*-Value
Forefoot R *	28.00 (2.00)	28.00 (5.00)	28.00 (5.00)	0.816
Mediofoot R *	8.00 (1.00)	7.00 (2.00)	8.00 (0.00)	0.191
Hindfoot R *	3.00 (1.00)	3.00 (2.00)	3.00 (1.00)	0.799
Wholefoot R *	39.00 (3.00)	40.00 (10.00)	40.00 (7.00)	0.971
Forefoot L *	29.00 (3.00)	28.00 (5.00)	27.00 (4.00)	0.676
Mediofoot L *	8.00 (1.00)	8.00 (1.00)	8.00 (1.00)	0.733
Hindfoot L *	3.00 (1.00)	3.00 (2.00)	3.00 (2.00)	0.915
Wholefoot L *	40.00 (6.00)	38.00 (9.00)	38.00 (7.00)	0.675

R = Right; L = Left; * Median (interquartile range), Kruskal–Wallis test.

**Table 2 ijerph-20-05033-t002:** Intragroup comparisons (T0 vs. T1) of plantar tactile sensitivity for the control group of older women participating in the study. Jequié, Bahia, 2022.

Variable	T0 (*Baseline*)	T1 (Post-Treatement)	*p*-Value
Forefoot R ^#^	27.60 (4.05)	27.07 (3.17)	0.505
Mediofoot R *	8.00 (1.00)	7.00 (2.00)	0.218
Hindfoot R ^#^	3.13 (0.99)	2.27 (0.79)	0.004
Wholefoot R ^#^	38.33 (5.76)	36.47 (4.71)	0.079
Forefoot L ^#^	28.00 (3.23)	27.13 (3.81)	0.155
Mediofoot L *	8.00 (1.00)	7.00 (3.00)	0.016
Hindfoot L *	3.00 (1.00)	2.00 (2.00)	0.005
Wholefoot L *	40.00 (6.00)	36.00 (10.00)	0.007

R = Right; L = Left; # Mean (standard deviation), Student’s *t* test for paired samples. * Median (interquartile range), Wilcoxon test.

**Table 3 ijerph-20-05033-t003:** Intragroup comparisons (T0 vs. T1) of the plantar tactile sensitivity for the conventional group of older women participating in the study. Jequié, Bahia, 2022.

Variable	T0 (*Baseline*)	T1 (Post-Treatement)	*p*-Value
Forefoot R *	28.00 (5.00)	29.00 (5.00)	0.095
Mediofoot R *	7.00 (2.00)	8.00 (1.00)	0.022
Hindfoot R *	3.00 (2.00)	4.00 (1.00)	0.026
Wholefoot R *	40.00 (10.00)	42.00 (7.00)	0.016
Forefoot L *	28.00 (5.00)	30.00 (5.00)	0.020
Mediofoot L ^#^	7.47 (1.19)	8.60 (1.35)	0.016
Hindfoot L ^#^	3.00 (1.13)	3.93 (0.96)	0.010
Wholefoot L *	38.00 (9.00)	42.00 (8.00)	0.007

R = Right; L = Left; # Mean (standard deviation), Student’s *t* test for paired samples. * Median (interquartile range), Wilcoxon test.

**Table 4 ijerph-20-05033-t004:** Intragroup comparisons (T0 vs. T1) of plantar tactile sensitivity for the exergame group of older women participating in the study. Jequié, Bahia, 2022.

Variable	T0 (*Baseline*)	T1 (Post-Treatement)	*p*-Value
Forefoot R *	28.00 (5.00)	30.00 (4.00)	0.003
Mediofoot R *	8.00 (0.00)	8.00 (1.00)	0.366
Hindfoot R ^#^	3.20 (0.86)	3.67 (0.81)	0.068
Wholefoot R ^#^	37.27 (6.12)	41.20 (5.14)	0.001
Forefoot L ^#^	27.07 (3.49)	29.13 (4.63)	0.021
Mediofoot L *	8.00 (1.00)	8.00 (2.00)	0.317
Hindfoot L ^#^	3.13 (0.99)	3.47 (1.25)	0.388
Wholefoot L ^#^	37.60 (5.04)	40.20 (6.63)	0.028

R = Right; L = Left; # Mean (standard deviation), Student’s *t* test for paired samples; * Median (interquartile range), Wilcoxon test.

**Table 5 ijerph-20-05033-t005:** Intergroup comparisons of changes (T1-T0) and effect size of the plantar tactile sensitivity of the older women participating in the study. Jequié, Bahia, 2022.

Variable	Control	Conventional	Exergame	*p*-Value	η^2^ Partial
Forefoot R *****	−1.00 (4.00) ^a^	1.00 (5.00) ^ab^	2.00 (6.00) ^b^	0.008	0.188
Mediofoot R *****	0.00 (2.00) ^a^	1.00 (2.00) ^b^	0.00 (1.00) ^ab^	0.032	0.141
Hindfoot R *****	−1.00 (1.00) ^a^	0.00 (1.00) ^b^	0.00 (1.00) ^b^	<0.001	0.279
Wholefoot R *****	−3.00 (4.00) ^a^	1.00 (5.00) ^b^	3.00 (5.00) ^b^	<0.001	0.278
Forefoot L *****	0.00 (3.00) ^a^	2.00 (2.00) ^b^	2.00 (6.00) ^b^	0.008	0.188
Mediofoot L *****	0.00 (1.00) ^a^	1.00 (2.00) ^b^	0.00 (1.00) ^ab^	0.001	0.249
Hindfoot L *****	−1.00 (1.00) ^a^	1.00 (2.00) ^b^	1.00 (2.00) ^b^	0.001	0.258
Wholefoot L *****	−3.00 (4.00) ^a^	4.00 (4.00) ^b^	2.00 (5.00) ^b^	<0.001	0.295

R = Right; L = Left; * Difference of medians T1-T0 (interquartile range), Kruskal-wallis test, Dunn’s post hoc; ^a,b^ Different letters in the lines indicate that the difference between groups was statistically significant (*p* ≤ 0.05); partial η^2^ = effect size.

## Data Availability

Research Support Foundation of the State of Bahia.

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
