# Peer review of "Analysis of Plantar Tactile Sensitivity in Older Women after Conventional Proprioceptive Training and Exergame"

_ijerph, 2023, doi:10.3390/ijerph20065033_

Round 1
Reviewer 1 Report (Previous Reviewer 3)
2.14.0.0This reviewed version of the manuscript improved substantially. Overall, the study is relevant to the field and promotes that different proprioceptive modalities are effective. The study is interesting, and they greatly improved the understanding of it by adding extra information in the results and methodology sections. However, I believe that the results are not categorical to indicate what they mention in their discussion, although there are differences, these are not significant, and they do not even discuss about it. The main focus of the study was to compare conventional methods with augmented or mixed reality systems. The main results are that there are significant improvements between the intervention and non-intervention group but, not really a discussion of the results of the intervention modalities, as it seems to be equally effective.
Section by section comments:
Introduction Section
1) What do “foot problems” specifically refer to? (line 35) it would be good to name a few. To then continue to focus on the “plantar tactile sensitivity” problem and its consequences, as authors do. This will enable to have a better context.
2) How is the problem of loss of plantar sensation normally treated or managed? What commercial or non-commercial platforms or systems exist? I would give examples. Although the authors mention that no studies compare them, this does not mean that no traditional therapy or technological elements are available on the market.
3) The introduction is very poor and vaguely introduces the topic of the research. If aging is the main aspect of the research, there should be a proper introduction in the literature to how aging affects skin proprioceptors and/or peripheral/central nervous system conditions that contribute to the problem. Emphasis should also be placed on orienting readers to why proprioceptive physical training works and the likely reasons why it helps to improve tactile sensitivity. Adding the Author’s response to this question in the introduction section greatly enhances the context and introduces the research topic.
Last author's reply, that it is not in the re-submitted version. Please add this part accordingly:
“In the somatosensory system, the peripheral nervous system can undergo changes such as loss of myelinated and unmyelinated fibers and a decrease in nerve conduction velocity, leading the elderly to have a deficit in sensory discrimination (UEDA; CARPES, 2013). According to Teixeira et al. (2011), with aging there is a loss of receptors and also a reduction in the number of sensory fibers that innervate the peripheral receptors, which may cause peripheral neuropathies that affect the proprioceptive system.
Proprioceptive rehabilitation is a resource widely used in physiotherapy in order to stimulate the sensorimotor system. In addition to offering the opportunity to experience in different situations and in an individualized way; encourage the individual's active participation, provide a motivating environment for learning, and facilitate the study of the characteristics of the participant's perceptual and motor skills and abilities. “
Methods Section:
4) This section improved significantly with the subsections. Further proper separation of the different modalities would improve readability as well.
5) The first appearance of the acronym TUGT (Timed Up & Go test) in line 68 should be defined. Also, the last T stands for "test" so it should be corrected to "TUGT" or " TUG test".
6) In the third paragraph of the "Population and sample" section. How are BMI High and Low groups defined? For reproducibility purposes of the experiment, numeric ranges for these groups should be clearly stated.
7) In the paragraph from lines 222 to 225, it is stated that monofilaments are presented by the thinnest until the perception is achieved. In the next sentence, it is stated that each monofilament was pressed into the skin. Please clarify if the test was done from low to high thickness until perception level or if each filament was presented and assessed in perception and localization. If the first sentence is true, then the second one should be corrected, indicating that "Each presented monofilament was pressed ...."
Results Section
8) Shapiro–Wilk test values should be shown for all feet tested regions. Observing results comparing sensitivity changes on the Right mediofoot region according to the median and Left mediofoot region, according to mean values. Stating that one side is normal and the other don’t, doesn’t seem to be correct, could be just a matter of sample sizes.
9) In Table 3, the footnote should be fixed as it states:" D=Right, E=Left " with "R=Right, L=Left" as all other tables.
10) The sentence in line 306, which reads "forefoot E" should be fixed to "forefoot L"
Discussion
11) Finally, I consider it a mistake to classify the intervention with exergames in this study as "virtual reality", since the subject is not totally abstracted from the real world. In this case, it is called augmented or mixed reality depending on the case. I propose changing the statements related to augmented reality technologies or games with motion monitoring, not virtual reality, as the authors do.
2.14.0.0
Author Response
All requests for corrections by reviewer 1 were met, as explained in the attached response letter.

Reviewer 2 Report (Previous Reviewer 1)
thank you for the reply from authors but reviewer still have one major query about the data analysis. Authors mention that "For this article it is not possible to repeat the analysis, but we will adopt this suggestion in another article." Please give more detailed explanation about why authors can't repeat the analysis or doing re-analyse? if not then authors need to mention this problem as limitation because this study will have problem with it's internal validity
Author Response
The question asked was answered in the attached response letter.

Round 2
Reviewer 2 Report (Previous Reviewer 1)
This paper now can be accepted in the current form. Authors already revised the manuscript according to the suggestion by all of the reviewers.
Author Response
Dear reviewer,
We thank you for the considerations previously sent, as we believe that they made our article better. We also thank you for approving the article.
A hug
This manuscript is a resubmission of an earlier submission. The following is a list of the peer review reports and author responses from that submission.
Round 1
Reviewer 1 Report
This is an interesting study but there is some queries need to resolved first by authors.
1. How do authors calculate the minimum sample size? because it seems the number of participants is quite not much in this study.
2. is there any reason why authors only have one evaluation after training?
3. Authors should use mixed regression model or GEE for this kind of analysis. Please recalculate your analysis using one of those model because this study had two group differences, between group difference (intervention vs control) and within group difference (T0 vs T1) which is not reliable only use difference compared between group.
Author Response
This is an interesting study, but there are some issues that need to be addressed first by the authors.
- How do the authors calculate the minimum sample size? because it seems that the number of participants is not much in this study.
As we did not have any studies published with this comparison, it was not possible to use a baseline study. Then, a sample calculation was performed based on the pilot study to validate this size.
Inserted a paragraph in the methodology after the second paragraph, to answer about the definition of the sample size.
- Is there any reason that authors have only one assessment after training?
Three evaluations were carried out after training, but for this article only the clipping of the first evaluation was included.
- Authors should use mixed regression model or GEE for this type of analysis. Please recalculate your analysis using one of these models because this study had two group differences, between group difference (intervention vs control) and within group difference (T0 vs T1), which is unreliable, just use compared difference between groups.
For this article it is not possible to repeat the analysis, but we will adopt this suggestion in another article.
Reviewer 2 Report
It is an original, innovative and highly interesting work. Provides new methods using current technology The abstract is adequate, and is divided into sections briefly summarizing the introduction, materials, results, and discussion. The introduction is appropriate, clear, well-founded, presents the objectives of the work, and does not present significant biases. The materials are described in a very detailed way and allow the necessary reproducibility for a work of these characteristics. However, the materials and methods are described continuously, throughout a text that is too long, without clear division into sections and making reading complex and difficult to understand. It is strongly recommended to divide the section into subtitles contemplating different aspects. The results are well presented, with well-organized tables, and do not present biases of importance. The data is objective and no bias, subjective comments, or unfounded data are detected. The discussion is slightly long, correct, it should be extended by adding limitations and advantages of the current manuscript, and comparison with other similar works not considered. The conclusions are adequate, clear and objective.
Author Response
The materials are described in a very detailed way and allow the necessary reproducibility for a work of these characteristics. However, the materials and methods are described continuously, throughout a very long text, without a clear division into sections, making the reading complex and difficult to understand. It is highly recommended to divide the section into subheadings covering different aspects.
The methodology was divided into subtopics to meet this request.
The discussion is a little long, correct, it should be expanded by adding limitations and advantages of the current manuscript, and comparison with other similar works not considered.
In the last two paragraphs of the discussion these questions are answered.
Reviewer 3 Report
The title is not matched between the submission platform and the manuscript. The title (on the manuscript) is not understandable since the “…Tactical….” and “…Planting…” terms are not correctly employed. Something relating to a comparison study on “plantar tactile sensitivity” in the elderly population between conventional and virtual reality training methodologies should provide a more direct meaning of what was done.
Abstract lines 25,26: “…had a better effect in relation to the outcomes studied…”. The significance between modalities and with respect to the control group should be stated here. It is not clear what “Better effect” means.
In the introduction, lines 33 to 52 state that the main goal is to observe changes in sensitivity due to aging factors. In Methods is further stated that comorbidities are excluded from the study. Introduction numbers (i.e., 80%) must be clarified further if it is due to aging factors only or to comorbidities and their prevalence (i.e., diabetes).
Esthesiometer (US English)/ Aesthesiometer (British English) is miss-spelled in the text. Line 167 “stesiometer”. Line 183“Estesiometer”.
The results start with underlying demographic data of patients, general age, and BMI are easy to understand and compare. Others, such as average income and education level, are not easily comparable and must be defined according to the specific population distribution. If there are some differences between people in this group with others, it must be stated how this segment behaves. Line 241, family income, should also be stated in USD, and what are the segments regarding specific populations. Line 241 also states the presence of disease; which diseases? Same with medications?
Foot regions were defined as forefoot, midfoot, hindfoot, and right and left whole foot regions. A figure segmenting the regions and test points for the examination would help the readers understand the involved zones rather than only descriptive locations from lines 177 to 182.
Since results are grouped in regions with different testing points, there should be a clear definition of what points are included in each group.
Labeling regions with D and E (assuming it stands for "esquerda e direita") should preferably be replaced by R and L or Right Left directly for clarity.
Lines 208 to 212 talk about the methodological aspect of evaluation with the sensitivity test.
The control group that did not undergo the training programs is not quite a controlled group. Even if it is stated, “The study excluded elderly women who had attended an- 62 other proprioceptive rehabilitation programs during the training,” normal daily living activity and behavior should be analyzed. (This isn't easy to assess and/or control in a long-time study like the one explained in the article over eight weeks.)
Between lines 213 and 231, statistical methods are described. It is not clear what the data distribution is. Since the beginning, paired Student's t-test or Wilcoxon test is mentioned. Is the data normally distributed? (Since it is required for Student's t-test). If the data is normal, there should be no difference between the mean and median. It’s then unclear why some results are shown regarding mean values and others regarding median.
Lines 304 to 307, states control group presented diminishing tactile sensitivity on both feet. What are the numbers supporting this? Can the variability in sensitivity scores be explained by worsening conditions on only two months? Or is it just the expected variability of the data from one test to the other? This should be further discussed rather than just stated upon a p-value.
Overall is difficult to distinguish the normal variability of the test measurements with only one measurement at the beginning and one at the end (eight weeks later).
Besides the tables, boxplots should be added, especially showing a comparison of scores pre and post-training for the different groups.
2.14.0.0 2.14.0.0Author Response
Dear reviewer,
Attached is the response letter, all requests for corrections have been met.

Round 2
Reviewer 3 Report
Overall the paper presents an 8-week follow-up study on a randomized control trial with a total cohort of 50 elderly women. The study aims to compare conventional and VR-based proprioception training to assess efficacy in reducing the aging effects of tactile sensitivity on the feet. According to the previously reported issues and the author's claims of correction. 1) Title issues. The new provided title: "ANALYSIS OF PLANTAR TACTILE SENSITIVITY IN ELDERLY WOMEN AFTER CONVENTIONAL PROPRIOCEPTIVE TRAINING AND EXERGAME." It reflects much better the presented work, so to this reviewer is ok. 2) Abstract lines 25,26: “...had a better effect in relation to the outcomes studied...”. The significance between modalities and in relation to the control group must be stated here. It is not clear what “better effect” means. The authors claim to have rewritten this paragraph but is not reflected in the resubmitted document.3) In the introduction, lines 33 to 52 state that the main objective is to observe changes in sensitivity due to aging factors. In Methods, it is also stated that comorbidities are excluded from the study. The numbers in the introduction (ie, 80%) should be further clarified if they are due solely to aging factors or comorbid conditions and their prevalence (ie, diabetes). The authors claim to have deleted a paragraph; it is unclear what paragraph was deleted as there seems to be no change in the resubmitted document. 4) The misspelled word "Esthesiometer" across the document seems to be corrected, so this reviewer accepts the change.
5) Results starting with underlying patient demographics, general age, and BMI are easy to understand and compare. Others, such as average income and education, are not easily comparable and must be defined according to the specific distribution of the population. If there are any differences between people in this group with others, it should be stated how this segment behaves. Line 261, family income, must also be declared in USD (with the currency equivalence at the time of evaluation), and which segments refer to specific populations. Line 261 also indicates the presence of disease; what diseases? Same with medications? The authors claim to have deleted paragraphs 1 and 2; they also refer to paragraph 14? Not clear what they point out. There seem to be no changes in the resubmitted document. 6) Foot regions were defined as forefoot, midfoot, rearfoot, and entire right and left foot regions. A figure segmenting the regions and test points for the exam would help readers understand the zones involved and their topological locations rather than just the descriptive locations of lines 177-182.
The authors were unable to fulfill this request. 7) As results are grouped into regions with different test points,
there should be a clear definition of which points are included in each group. The authors claim this information is already in the text in paragraph lines 215-222, which is not the case, because there only the amount of points per region is declared, not which test points are referring to. The issue is that first, the test points are declared in regions: plantar region (PR), Medial Region (MR), and Interphalangeal Region (IR). Then the paper refers to forefoot, midfoot, rearfoot, and whole foot regions. The information is unclear, and with the requested figure in point 6, it would be clearer and easier to reproduce in future works.
8) Labeling regions with D and E (assuming it means "left and right") should preferably be replaced with R and L or Right Left directly for clarity. Even if tables caption designs D to represent R and E to represent L, the language should be fixed to be more easily interpreted.
The authors claim to fix this. There seem to be no changes in the resubmitted document. 9) Lines 223 to 227 talk about the methodological aspect of the evaluation with the sensitivity test. The control group that did not go through the training programs is not exactly a controlled group. Even if it says: “The study excluded elderly women who attended another proprioceptive rehabilitation program during training,” normal daily living activity and behavior should be analyzed. (This is not easy to assess and/or control in a long-term study like the one explained in the article over eight weeks.) The authors claim: "This paragraph was included to answer this question: It should be noted that the elderly women were rigorously monitored over the 8 weeks, noting their non-participation in activities that could influence the study." However, there seem to be no changes in the resubmitted document. 10) Regarding the previous point, assessment of daily living activities should be assessed since comparison of the elderly population with different daily activities will furthermore add variability in data and for sure impact the baseline and control group. This must be assessed at least qualitatively using a standard scale.
11) Between lines 230 and 236, statistical methods are described. It is not clear what the distribution of the data is. The paired Student's t-test or the Wilcoxon test is mentioned from the beginning. Is the data normally distributed? (Since it is required for the Student's t-test). If the data are normal, there should be no difference between the mean and median. Thus, it is unclear why some results are shown in relation to mean values and others in relation to the median. For this reviewer is still not clear the ways data was analyzed. Further emphasis on the next points as well. The authors claim: Below each table, there is information on the variables that had a normal distribution or not, so averages and medians and their specific tests were highlighted.
12) In lines 324 to 327, the control group of states showed decreased tactile sensitivity in both feet. What are the numbers that prove this? Can the variability in sensitivity scores be explained by conditions worsening in just two months? Or is it just the expected variability of the data from one test to the next? This should be discussed more rather than just stated about a p-value. The authors claim: Results were found that showed worsening in this control group, but a more specific study would be needed to control more confounding variables that could answer the reason for this worsening, which in this case is beyond the objective of this study. The reviewer agrees that explaining the aging factors underlying the worsening is out of scope. However, the amount of data presented in the article seems statistically insufficient to claim this point, especially if only one measurement was taken before and after the evaluation time since there is no information on how variable the sensitivity score of the applied test is and its distribution.
13) It is often difficult to distinguish normal variability from test measurements, with only one measurement at baseline and another at the end (eight weeks later). In addition to the tables, boxplots should be added, mainly showing the comparison of pre- and post- training scores for the different groups.
The authors claim: For this article, it will not be possible, as the authors do not work with boxplots. For this reviewer, this is not a feasible answer, and clarifying the used data distribution that underwent the statistical analyses is critical.
14) The article deals with multiple experiment cohorts and multiple time measurement intervals for inter subjects analysis. Linear mixed models (LMM) should be used and state the difference between fixed and random effects on data.
15) The results are not clear in the way it is presented. After all, why tables mixes mean values and median values is unclear. For example, Table 3 displays the statistical results of scores before and after the conventional training cohort. The Right Mediofoot region is presented regarding its median score value for the group. While the Left Mediofoot region is presented with its mean score value. 16) Introduction is very poor and vaguely introduces the research topic. If aging is the main research aspect, there should be a properly introduced in the literature of how aging affects skin proprioceptors and/or peripheral/central nervous system conditions contributing to the problem. Emphasis should also be taken to guide readers on why physical proprioceptive training works and probable reasons how it helps improves tactile sensitivity. 17) introduction sentences like "In this regard, for every three elderly people living in the community, at least one of them has foot problems, which can affect 80% of the elderly population, especially women, who have about twice as many foot problems classified as moderate and severe [3]" Leads to confusion, since the problem is presented on 1/3 of elderly people (33%)or 80% is affected. Authors should be specific about what they refer to and the prevalence of each aspect of what they are presenting. 2.14.0.0aSF